# Transfer learning for non-image data in clinical research: A scoping review

**Andreas Ebbehoj** [1,2]☯, **Mette Østergaard Thunbo** [2]☯, **Ole Emil Andersen** [3], **Michala Vilstrup Glindtvad** [3], **Adam Hulman** [3]*

**1** Department of Endocrinology and Internal Medicine, Aarhus University Hospital, Denmark, **2** Department of Clinical Medicine, Aarhus University, Denmark, **3** Steno Diabetes Center Aarhus, Aarhus University Hospital, Denmark

☯ These authors contributed equally to this work.
* adahul@rm.dk

## Abstract

### Background

Transfer learning is a form of machine learning where a pre-trained model trained on a specific task is reused as a starting point and tailored to another task in a different dataset. While transfer learning has garnered considerable attention in medical image analysis, its use for clinical non-image data is not well studied. Therefore, the objective of this scoping review was to explore the use of transfer learning for non-image data in the clinical literature.

### Methods and findings

We systematically searched medical databases (PubMed, EMBASE, CINAHL) for peer-reviewed clinical studies that used transfer learning on human non-image data.

We included 83 studies in the review. More than half of the studies (63%) were published within 12 months of the search. Transfer learning was most often applied to time series data (61%), followed by tabular data (18%), audio (12%) and text (8%). Thirty-three (40%) studies applied an image-based model to non-image data after transforming data into images (e.g. spectrograms). Twenty-nine (35%) studies did not have any authors with a health-related affiliation. Many studies used publicly available datasets (66%) and models (49%), but fewer shared their code (27%).

### Conclusions

In this scoping review, we have described current trends in the use of transfer learning for non-image data in the clinical literature. We found that the use of transfer learning has grown rapidly within the last few years. We have identified studies and demonstrated the potential of transfer learning in clinical research in a wide range of medical specialties. More interdisciplinary collaborations and the wider adaption of reproducible research principles are needed to increase the impact of transfer learning in clinical research.

**Data Availability Statement:** The data extracted from the identified articles and then presented in the Results section is available as an electronic supplement along with the code used for the analysis.

**Funding:** OEA, MGV and AH are employed at Steno Diabetes Center Aarhus that is partly funded by a donation from the Novo Nordisk Foundation. AE and MØT are supported by PhD scholarships from Aarhus University. The funders had no role in study design, data collection and analysis, decision to publish, or preparation of the manuscript.

**Competing interests:** The authors have declared that no competing interests exist.

# Introduction

There is no doubt that most clinicians will use technologies integrating artificial intelligence (AI) to automate routine clinical tasks in the future. In recent years, the U.S. Food and Drug Administration has been approving an increasing number of AI-based solutions, dominated by deep learning algorithms [1]. Examples include atrial fibrillation detection via smart watches, diagnosis of diabetic retinopathy based on fundus photographs, and other tasks involving pattern recognition [1]. In the past, the development of such algorithms would have taken an enormous effort, both regarding computational capacity and technical expertise. Nowadays, computational tools, including cloud computing, free software, and training materials are more easily accessible than ever before [2]. This means that more researchers, with more diverse backgrounds, have access to machine learning, and that they can focus more on the subject matter when developing AI-based solutions for clinical practice. Despite this trend, machine learning, neural networks, transfer learning, and other elements of AI still seem to be surrounded by mystery in the clinical research community, and AI has yet to reach its potential in the clinic [1].

A neural network is a type of machine learning model, inspired by the structure of the human brain (S1 Glossary). In the simplest scenario, the input data flows through layers of artificial neurons, known as hidden layers. Each hidden neuron takes the results from previous neurons, calculates a weighted sum before applying a nonlinear function, and feeding this value forward to the next layer. The final layer transforms the results according to the prediction task, for example to probabilities, when the task is to predict whether a lung nodule on a chest X-ray is malignant. Fitting or training a neural network means optimizing the weights or parameters against some performance metric. Deep learning means the use of neural networks with several hidden layers of neurons. In the last decade, several neural network architectures were designed with some consisting of more than 100 layers and tens of millions of weights [3]. In such a deep neural network, neurons at lower levels (i.e. closer to input layer) 'learn' to recognize some lower-level features in the data (e.g. circles, vertical and horizontal lines in images), which the higher level neurons combine into more complex features (e.g. a face, some text, etc. in an image) [4]. Neural networks are popular tools in machine learning as they can approximate any complex nonlinear association. This flexibility, however, comes with a price, as fitting complex neural networks requires very large datasets, which limits the spectrum of fields where their application is feasible.

Transfer learning circumvents the above-mentioned limitation and unlocks the potential of machine learning for smaller datasets by reusing a pre-trained neural network built for a specific task, typically on a very large dataset, on another dataset and potentially for a different task. The pre-trained model is also known as the source model, while the new dataset and task is referred to as the target data and target task. A common example is to take a computer vision model, trained to identify everyday objects in millions of images, and further train this model for grading diabetic retinopathy on only a few thousand fundus photographs, instead of training the model from scratch on this smaller dataset [5]. This example demonstrates a type of transfer learning known as fine-tuning, or weight or parameter transfer. Another type of transfer learning is feature-representation transfer, where the features from the hidden layers of a model are used as inputs for another model, but other forms of transfer learning exist [6–8].

Applications of transfer learning are common in computer vision, where large datasets are publicly available to train models that can then be adapted to different domains. One of the most influential datasets is ImageNet, which includes more than a million images from everyday life [9]. Two recent scoping reviews focused on transfer learning for medical image analysis, one of which identified around a hundred articles applying ImageNet-based models for

clinical prediction tasks [10, 11]. The number of published articles using transfer learning for medical image analysis approximately doubled every year in the last decade, demonstrating an increasing interest in transfer learning [10, 11]. To our knowledge, a comprehensive overview of the use of transfer learning for other non-image data types is lacking, despite that tabular and time series data seem to dominate in the clinical literature.

Therefore, the objective of this scoping review was to fill this knowledge gap by exploring and characterizing studies that used transfer learning for non-image data in clinical research.

## Methods

Scoping reviews identify available evidence, examine research practices, and characterize attributes related to a concept (i.e. transfer learning in our case) [12]. This format fits better with our research objective than a traditional systematic review and meta-analysis, as the latter require a more well-defined research question. During the process, we followed the 'PRISMA for Scoping Reviews' guidelines and the manual for conducting scoping reviews by the Joanna Briggs Institute [13, 14].

### Eligibility criteria

In accordance with the aim of this review, we only wanted to include studies using transfer learning for a clinical purpose. Similarly, we were only interested in articles indexed in medical databases, as opposed to in purely technical databases such as The Institute of Electrical and Electronics Engineers (IEEE) Xplore Digital Library, which most clinical researchers and practitioners might be unfamiliar with. Moreover, such articles are often written in a technical language, which limits their utility for the clinical research community. By only including articles indexed in medical databases, we could gauge how exposed clinical researchers are to clinical use of transfer learning and could also provide a list of articles that can serve as inspiration to clinical researchers interested in transfer learning.

To be considered for inclusion, studies needed to 1) be published, peer-reviewed, written in English, and indexed in a medical database (defined as PubMed, EMBASE, or CINAHL) since database inception, 2) be a clinical study or focus on clinically relevant outcomes or measurements, 3) use data from human participants or synthetic data representing human participants as target data, 4) use transfer learning with either fine-tuning (parameter transfer) or feature-representation transfer, and 5) analyze non-image target data (text, time-series, tabular data, or audio). Accordingly, we excluded preprints and conference abstracts, basic research, cell studies, animal studies, and studies analyzing image data. Videos were considered as blends of audio and images; therefore, we did not include studies analyzing this type of target data. However, we did not exclude studies that converted non-image data into images (e.g. converting an audio file into an audiogram) and then analyzed them using models from computer vision. Finally, we also excluded studies which combined source data with the target data as an integral part of their analysis as opposed to reusing only a source model. Eligibility criteria were predefined and the review protocol is available online [15].

### Information sources and search strategy

We searched PubMed, EMBASE, and CINAHL from database inception until May 18, 2021. The search strategy was drafted by the authors with the assistance from an experienced librarian. The final search strategies for all databases are available online [15]. In brief, we searched for all studies that either specifically mentioned 'transfer learning' or included both a similar but less specific phrase (e.g. 'transfer of learning', 'transfer weights', 'connectionist network', etc.) and an AI-related keyword (e.g. 'artificial intelligence', 'neural network', 'NLP', etc.). The

search strategy was supplemented by a call-out on Twitter by AH (@adamhulman) on May 25, 2021, and by scanning the references of relevant reviews found during the screening process. As per our aim, we did not include grey literature.

## Selection of sources of evidence

The search results from each database were imported to the reference program Endnote 20.1 (Clarivate Analytics, Philadelphia, PA, USA). Duplicate removal was done by AE and is documented online [15]. Hereafter, the records were transferred to Covidence (Veritas Health Innovation, Melbourne, Australia) for the screening process.

Abstracts and titles were screened against the predefined eligibility criteria by at least two independent authors (AE, MØT, OEA, AH). If an abstract and title clearly did not meet the inclusion criteria, the record was excluded. In case of uncertainty, the record was included for full-text screening. Next, full-text versions of all reports were retrieved if possible and assessed for eligibility. If a report was excluded at this step of the screening process, a reason was documented. In any stage of the screening process, conflicts were resolved through discussion between the dissenting authors. In case of no consensus, the last author (AH) made the final decision. The study selection process is presented using a PRISMA flow diagram [16] in Fig 1.

## Data charting process

Research questions were pre-specified and published in the scoping review protocol [15]. Data of interest from each included study was extracted by two independent researchers based on a

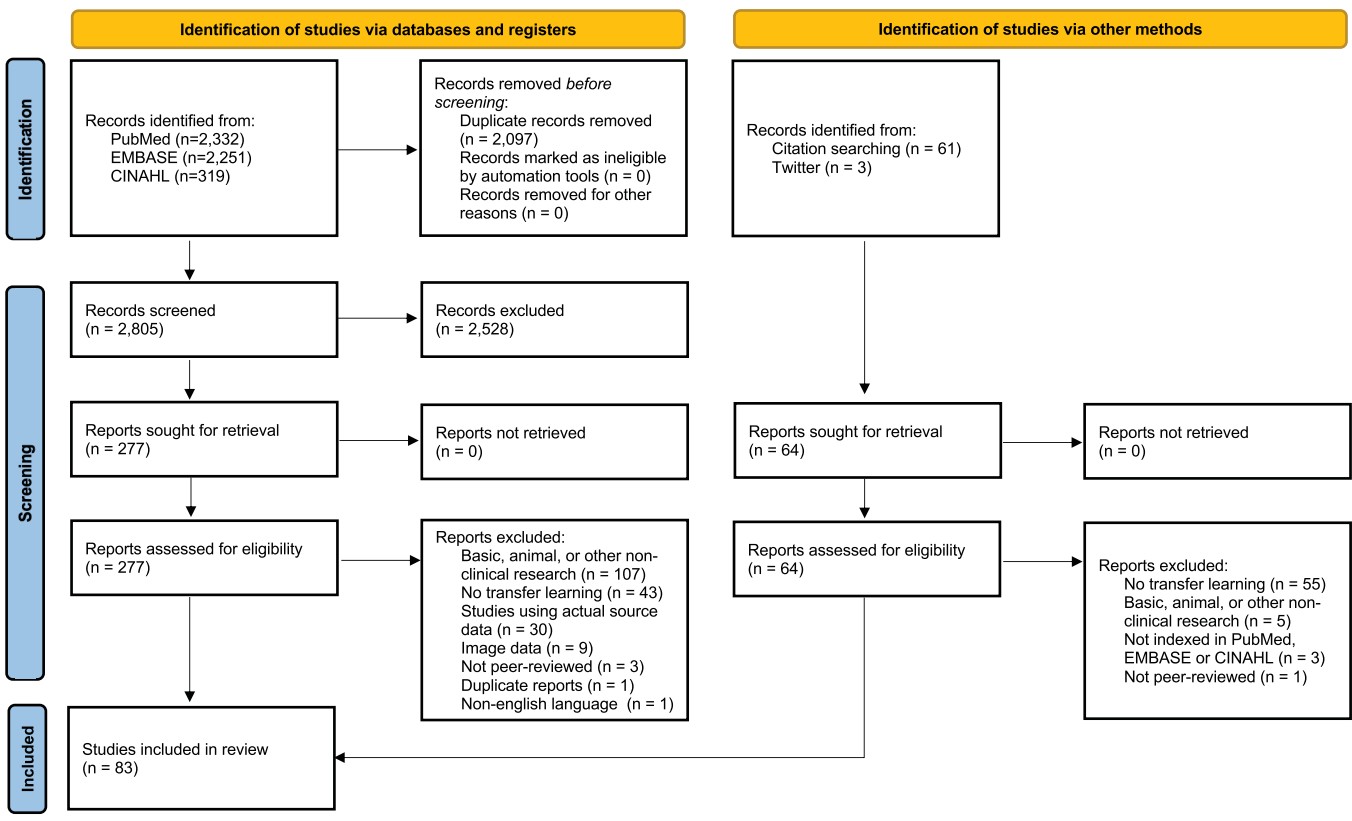

**Fig 1. PRISMA flow diagram [16].**

**Table 1. Included articles by data type and primary medical specialty.**

| | Time series | Tabular | Audio | Text |
|---|---|---|---|---|
| Neurology | Abou Jaoude 2020, Andreotti 2018, Artoni 2020, Banluesombatkul 2020, Chambon 2018, Daoud 2019, Gao 2020, Hssayeni 2021, Ilakiyaselvan 2020, Jadhav 2020, Li, Q. 2021, Narin 2020, Nasseri 2021, Nogay 2020, Phan 2020, Phan 2020, Raghu 2020, Wang 2019, Yan 2021, Yang 2021, Zhang, B. 2020, Zhang, H. 2020 | | Balagopalan 2021, Chlasta 2020, Nodera 2019, Roshanzamir 2021 | |
| Cardiology | Abdelazez 2020, Al Rahhal 2018, Allen 2021, Ghaffari 2019, Jang 2021, Jiang 2019, Lopes 2021, Naz 2021, Shi 2020, Shyam 2019, Strodthoff 2021, Tadesse 2019, Torres-Soto 2020, Weimann 2021, Yang 2020, Yin 2019 | | Koike 2020, Tseng 2021 | |
| Infectious diseases | Li, Y. 2021, Tadesse 2020, Xu 2020 | Wardi 2021 | Imran 2020 | |
| Psychiatry | Sadouk 2018, Shalbaf 2020 | | | Dai 2020, Du 2018, Howard 2020 |
| Genetics | | AlShibli 2019, Gao 2020, Kim 2020, López-García 2020, Qiu 2020 | | |
| Endocrinology | De Bois 2021, Kushner 2020, Yildirim 2019 | | | |
| Pulmonology | | Bae 2021 | Demir 2020, Hsiao 2020 | |
| Epidemiology | Song 2021 | | | Li, Y. 2020, Si 2020 |
| Pathology | | Mostavi 2021, Santilli 2021, Seddiki 2020 | | |
| Pharmacology | | Ekpenyong 2021 | | Al-Garadi 2021 |
| Otorhinolaryngology | Olsen 2020 | | Luo 2020 | |
| Neonatology | O'Shea 2021 | He 2020 | | |
| Intensive care | | Shickel 2021, Steinberg 2021 | | |
| Surgery | | Chia 2012 | | |
| Geriatrics | Martinez 2020 | | | |
| Oncology | | | | Syed 2020 |
| Gastroenterology | Agrusa 2020 | | | |

Some articles could have been assigned to more medical specialties, but the authors have chosen a primary one with consensus for the sake of simplicity.

pre-developed and tested data extraction form [15]. Disagreements were solved as described above. Data on study characteristics, like the study area within medicine, the affiliation of the authors (medical or technical departments), and the aim of the study, was extracted. Furthermore, extracted data included knowledge on model characteristics such as what method and origin the model being transferred is based on, type of transfer learning, type of source and target data, and the advantages/disadvantages if compared to a non-transfer learning method. Lastly, information on the reproducibility of the studies was registered, i.e. the public availability of the data, the reused model, and the code for the analysis, as well as the software used. The studies are listed by field within medicine and data type in Table 1, and the complete dataset is available in the electronic supplement (S1 Data).

## Statistical analysis

Categorical variables were described with frequencies and percentages. The different combinations of source and target data were visualized using a Sankey diagram. Analyses were conducted in R 4.1.0 (R Foundation for Statistical Computing, Vienna, Austria) and Stata

Statistical Software, Release 16.1 (StataCorp, College Station, Texas) was used for data management. The R code and the results are available as an RMarkdown document in the electronic supplement (S1 Code).

## Results

The search resulted in 4,902 records, of which 2,097 were duplicates (Fig 1). After screening the remaining 2,805 records, 2,528 were excluded as irrelevant. Of the 277 records included for full-text review, 194 were excluded, mainly because they focused on basic, animal, or other non-clinical research (n = 107) or did not use transfer learning (n = 43). Another 64 records were identified in reviews or on Twitter, but none of the papers were eligible for inclusion. In total, 83 studies were included in this review (Table 1).

Only one of the identified studies [17] was published before 2018 (in 2012) and 52 out of 83 (63%) studies were published in the 12 months preceding the search conducted on 18th of May, 2021 (Fig 2). Seventeen (20%) studies were published in journals or proceedings of IEEE.

The most common fields of studies were neurology (n = 26) and cardiology (n = 18) followed by genetics, infectious diseases and psychiatry (n = 5 for each). In line with the medical specialties, analyses of electroencephalography (EEG) and electrocardiogram (ECG) data were common with 20 and 19 studies, respectively. Study aims were most often described with the terms: 'prediction', 'detection', and 'classification'.

In total, 50 out of 83 (60%) studies included at least one author with a clinical affiliation and at least one with a technical affiliation. Studies with pure technical affiliations were more common (29 out of 83; 35%) than studies with pure clinical affiliations (4 out of 83; 5%), except for studies analyzing text, where all articles were written by clinicians with or without technical co-authors.

Time series was the most common target data type (n = 51; 61%), followed by tabular data (n = 15; 18%), audio (n = 10; 12%) and text (n = 7; 8%). The reused model was developed in a source dataset of a different type from the target data in 36 out of 83 (43%) of the studies (e.g. transforming time series to spectrograms before applying an image-based model). Transformation before analysis was often applied to time series (25 out of 51; 49%), while it was not used in the analysis of text at all. Image-based models were reused in 33 out of 83 (40%) studies, dominated by models developed in the ImageNet [9] dataset (n = 27). Fig 3 shows the different combinations of source and target data types.

Fine-tuning was approximately three times as common (58 out of 83; 70%) as feature-representation transfer (18 out of 83; 22%). Seven studies applied both approaches. Convolutional neural networks were the most often reused models (58 out of 83; 70%), followed by recurrent neural networks (18 out of 83; 22%). Among the 25 studies applying feature-representation transfer, support vector machines and 'vanilla' neural networks were the most common methods used as the final model producing outputs based on the feature representations. Seven studies applied and compared multiple methods for this purpose [18–24].

Half of the studies (41 out of 83; 49%) reused a publicly available model. Among these, 33 out of 41 (80%) models were from an external source, while the rest 20% developed the reused model as part of the actual study and made it publicly available.

Python was the most used software (43 out of 83; 52%), followed by MATLAB (12 out of 83; 14%), while the rest of the articles did not mention which software they used for the analysis. Overall, 22 out of 83 (27%) studies made their analysis code publicly available, and 55 out of 83 (66%) studies utilized at least one open access dataset. The PhysioNet repository [25] was mentioned as a data source in 25 out of 55 (45%) of these articles.

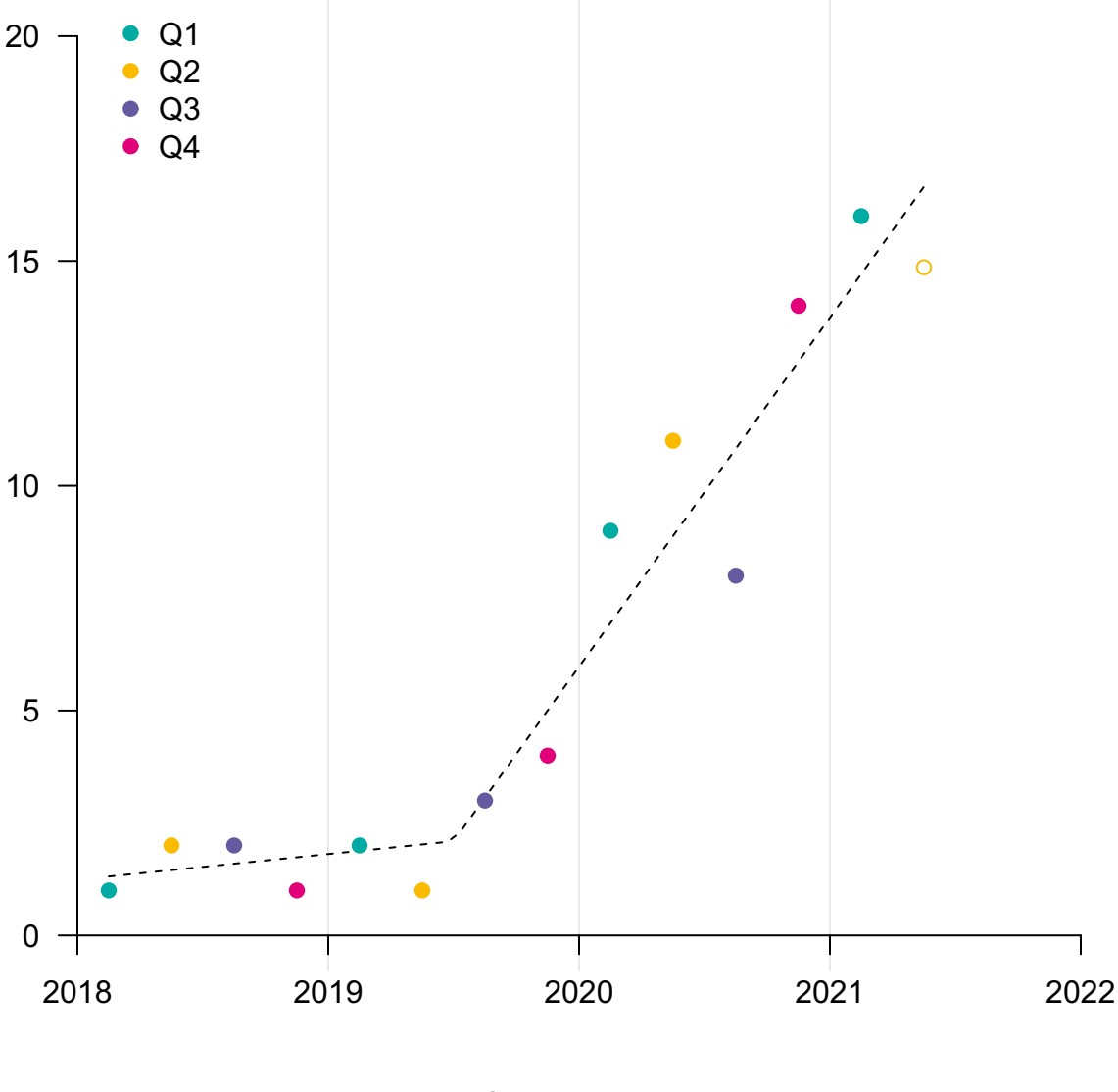

**Fig 2. Number of articles published quarterly since 2018.** One article [17] published in 2012 is not shown on this graph. 2021/Q2 is an extrapolated value based on the number of articles in the observation period (01/03/21-18/05/21).

## Discussion

Applications of transfer learning were identified in a variety of clinical studies and demonstrated the potential in reusing models across different prediction tasks, data types, and even species. Improvements in predictive performance were especially striking when transfer learning was applied to smaller datasets, as compared to training machine learning algorithms from scratch. Image-based models seemed to have a large impact outside of computer vision and were reused for the analysis of time series, audio, and tabular data. Many studies utilized publicly available datasets and models, but surprisingly few shared their code.

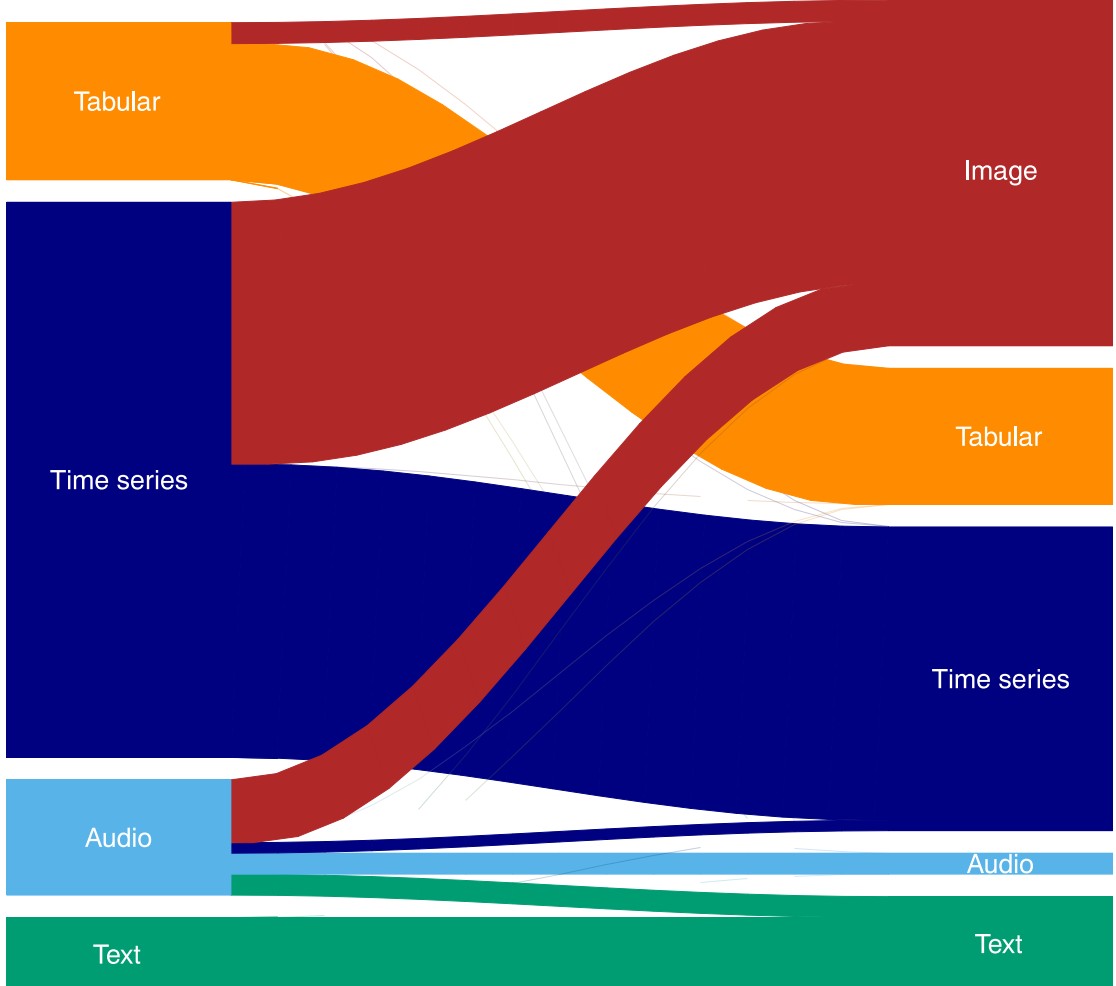

**Fig 3.** Visualization of data types (target) analyzed in the studies (on the left side) and the type of data that the reused model (source) was built on (on the right side).

Transfer learning via fine-tuning and feature-representation learning has almost been unknown in the clinical literature until 2019, when the number of articles started to increase rapidly. This development is lagging a few years behind the trend seen in medical image analysis [11], which might be explained partly by the impact that computer vision models have on non-image applications too. Also, we only included peer-reviewed articles indexed in medical databases, while the latest developments in computer science are most likely to be first published as preprints or conference proceedings.

More than half of the studies were from the fields of neurology and cardiology, while the rest of the studies were more equally distributed across other fields. This could be due to the high-resolution nature of EEG and ECG data that suits well with the application of machine learning. Also, there are many publicly available datasets and data science competitions that attract the attention of the computer science community.

The majority of study aims were predictions of binary or categorical outcomes on the individual level e.g. detection of a disease, classification of disease stages, or risk estimation. Few studies focused on prediction of continuous outcomes (e.g. glucose levels [26, 27]) and forecasts on the population level (e.g. COVID-19 trend [28] and dengue fever outbreaks [29]).

Prediction models are abundant in the clinical literature, however most do not make it into routine clinical use [30]. For machine learning algorithms, one of the barriers is that clinical practitioners often consider them as black boxes without intuitive interpretations, even though new solutions are constantly being developed to solve this problem [31]. Another issue is that machine learning algorithms are often used to analyze modestly sized tabular datasets and (not surprisingly) fail to outperform traditional statistical methods, contributing to even more skepticism. Transfer learning helps to overcome this problem by reusing models trained on large datasets and tailoring them to smaller ones. This provides the opportunity to capitalize on recent developments in machine learning research in new areas of clinical research. However, to guarantee the development of easily accessible and clinically relevant models that go beyond the 'proof-of-concept' level into clinical deployment, it is crucial that computer scientists and clinical researchers work together, which we observed in less than two-thirds of the studies. This proportion would probably have been even lower if we had considered studies indexed in non-clinical databases. Even the clinical studies using transfer learning, we identified, are still largely being published in rather specialized journals focusing on an audience with some technical background (e.g. IEEE and interdisciplinary journals) and have not yet reached mainstream clinical journals.

We were surprised by the high number of studies reusing a model that was developed in a dataset of a different data type than the target dataset. This also made it difficult to compare the size of source and target datasets quantitatively, as they might have been characterized in very different units (e.g. hours of audio recordings in the target dataset vs. number of images in the source dataset), but our observation was that the size of the target datasets was often much smaller than the source dataset. E.g. models developed in the ImageNet dataset (>1 million images) were used even in smaller clinical studies with <100 patients [24, 32, 33], where the use of machine learning models would otherwise not be recommended or feasible.

Many studies compared long lists of different model architectures (e.g. ResNet [34], Inception [35], GoogLeNet [36], AlexNet [37], VGGNet [38], MobileNet [39] etc.) or even different methods. This, often combined with the use of several performance metrics (area under the ROC-curve, F1-score, accuracy, sensitivity, specificity, etc.), makes it difficult to summarize the advantages/disadvantages of transfer learning, even though many studies included comparisons with other deep learning solutions without transfer learning. We observed that the degree of performance improvements varied greatly. Some studies reported faster and more stable model fitting process with transfer learning due to better utilization of data. Some of these results are described in the sections below characterizing the different data types.

Even though half of the studies reused a publicly available model and even more used a publicly available dataset, principles of reproducible research were followed less often. One-third of the studies did not report at all which software they used and even fewer shared their code. Those who did, most often used Python, then MATLAB, while we did not find a single study using R for the transfer learning part. This might be a consequence of the fact that commonly used libraries like PyTorch, TensorFlow, or fastai, are native in Python and only recently became available in R. Those authors who shared their code did so almost exclusively on GitHub.

A discussion of data type-specific observations is organized in the following four sections.

## Time series

Time series was the most common target data type among the included studies, dominated by studies of sleep staging and seizure detection based on EEG data [19, 32, 40–55] and ECG analyses [23, 24, 56–70] (e.g. arrhythmia classification). Moreover, transfer learning was used for

prediction of glucose levels [26, 27], estimation of Parkinson's disease severity [71], detection of cognitive impairment [33] and schizophrenia [52], and forecasting of infectious disease trends [28] and outbreaks [29], among other applications [72–82].

Almost half of the studies transformed their data into images (e.g. spectrograms, scalograms), most often using Fourier (e.g. short-time or fast) or continuous wavelet transforms, to be able to utilize models from computer vision. A reason for this might be that publicly available time series datasets were rather small until recently, while in computer vision, ImageNet has been described and widely used as a large benchmark dataset for about a decade [9]. In 2020, PTB-XL [83] was described as the largest publicly available annotated clinical ECG-waveform dataset including 21,837 10-second recordings from almost twenty thousand patients. Recently, Strodthoff et al. [64] used this dataset for ECG analysis, in a similar role to ImageNet's for computer vision and provided benchmark results for a variety of tasks and deep learning methods. The authors also demonstrated that a transfer learning solution based on PTB-XL provided much more stable performance in another smaller dataset when the size of the training data was further restricted in an experiment, as compared to models developed from scratch.

Transfer learning can enable the use of machine learning models in small datasets, where it would not be feasible to train models from scratch. This may have a positive impact on the study of rare diseases or minority groups, regardless of the type of data. Lopes et al. [69] used this strategy to detect a rare genetic heart disease based on ECG recordings, although they only had 155 recordings from patients. First, the authors developed a convolutional neural network for sex detection, which required a readily available outcome from their database including approximately a quarter of a million ECG recordings. Even though the model was initially trained for a task with little clinical relevance, the model still learnt some useful features from the data. This model was then fine-tuned for the detection of a rare genetic heart disease using only 310 recordings from 155 patients and 155 matched controls. With this approach, the authors achieved major improvements in model discrimination not only compared to training the same architecture from scratch, but also outperformed various machine learning approaches and clinical experts [61].

In the previous example, an easily accessible outcome or label (i.e. sex) made it possible to train the base model, however even this can be avoided by using autoencoders [84]. An autoencoder is an unsupervised machine learning algorithm often used for denoising a dataset by first reducing its dimensions and only keeping relevant information (encoding), before reconstructing the original dataset (decoding). The feature representations learnt during the encoding step might be then transferred to new tasks. Jang et al. [59] developed an autoencoder based on 2.6 million unlabeled ECG recordings which was then reused as the base of an ECG rhythm classifier. The authors compared this approach to an image-based transfer learning classifier and a model trained from scratch. The autoencoder solution performed best, but the difference to the model trained from scratch was much smaller than in the other example. More interestingly, the authors compared the results when using 100%, 50% and 25% of the available data, and found a major drop in performance of the model trained from scratch when using only 25% of the data, while the autoencoder-based transfer learning solution still had an excellent performance as it still indirectly utilized data from >2 million ECG recordings. The image-based solution had a slightly lower F1-score (indicating worse performance) than other methods when using 100% of the data, but it did not change much when using only 25% of the data, which led to a better performance than of the model trained from scratch.

EEG data were most often used for sleep staging and epileptic seizure detection. We highlight the study by Raghu et al. [51] using pre-trained convolutional neural networks for seizure detection based on EEG-based spectrograms, because the authors described both fine-tuning

and feature-representation learning solutions. In the latter approach, features were fed into a support vector machine classifier, a popular machine learning technique. The authors found that taking features from deeper layers, representing higher-level features, resulted in higher accuracy for most architectures. For almost all architectures, feature-representation transfer was found to be better than fine-tuning regarding accuracy. However, this can partly be a consequence of the fact that many more models were fitted including features from different layers. Further, once the authors found out which layer provided the most useful representation for a specific problem, the optimization process was shortened markedly compared to the fine-tuning process (~4 vs. 52 min with the Inception-v3 architecture).

## Audio

Voice recognition and other audio-based applications of AI surround us in our everyday lives. In a clinical setting, doctors have traditionally used audio signals from stethoscopes to screen patients e.g. for heart and lung diseases. Additionally, vocal biomarkers processed with machine learning algorithms are getting more and more attention in a research setting, and are expected to aid diagnosis and monitoring of diseases in the future [85]. Despite the increasing interest and opportunities for relatively easy and cheap data collection, we only found 10 studies that used transfer learning on audio data. However, these studies covered a variety of fields in medicine (and corresponding audio signals): neurology (speech and electromyography [21, 22, 86, 87]), cardiology (heart sound [88, 89]), pulmonology (respiratory sounds [90, 91]), infectious diseases (cough [92]), and otorhinolaryngology (breathing [93]).

Recordings of speech contain different levels of information: linguistic features that can be derived from transcripts, and temporal and acoustic features that can be derived from raw audio recordings. Two studies applied BERT [94], an open source, pre-trained natural language model, to analyze transcripts of speech with the aim of diagnosing Alzheimer's disease [21, 87]. Balagopalan et al. [87] also tested a model for the same task that included acoustic features, however, this model performed worse than the model only based on linguistic features. This finding highlights the possible complexity of audio data analysis and the importance of benchmarking different models against one another to get the best performing algorithm.

Machine learning algorithms pre-trained on labeled image-data, like ImageNet, can recognize features on non-image data like audio [95]. To use image models on audio data, the data has to be transformed, usually into a spectrogram image. We were surprised to find that half of the audio studies used pre-trained image models to analyze audio data, and only two studies used pre-trained audio models. This, again, highlights the impact of image models even outside of computer vision. Of interest, Koike et al. [88] aimed to predict heart disease from heart sounds with transfer learning and compared two models: one trained on audio data and another on images. They found the transfer learning model pre-trained on the open source sound database, AudioSet [96], performed better than image models of different architectures such as VGG, ResNet, and MobileNet. Open source, labeled audio datasets exist (e.g. AudioSet [96], LibriSpeech [97]), and the results of Koike et. al highlight how models pre-trained on audio might perform better than image models.

With audio data easily obtainable in a clinical setting, there is a great opportunity to develop and improve transfer learning models that can aid clinicians to screen and diagnose patients in a cost-effective way.

## Tabular data

Tabular data is probably the most used data type within clinical research. However, we only identified 15 studies using transfer learning on tabular data covering very different fields in

medicine: two-thirds of them were from genetics [98–102], pathology [103–105], and intensive care [18, 106], while the remaining five were from surgery [17], neonatology [107], infectious disease [108], pulmonology [109], and pharmacology [110]. Oncological applications like classification of cancer and prediction of cancer survival were common among the studies in genetics or pathology. As an example for zooming in from a broader disease category to a specific, rarer disease, one study developed a model using gene expression data from a broad variety of cancer types and then fine-tuned the model using a much smaller dataset to predict cancer survival in lung cancer patients [101].

Transformation of tabular data into images was rare compared to time series and audio data. We only identified two studies transforming gene expression data into images [98, 101] and then applying computer vision models for a prediction task. Both of these studies used open access gene expression datasets: AlShibili et al. [98] studied classification of cancer types in a dataset from cBioPortal [111], and López-García et al. [101] predicted cancer survival in a dataset from the UCSC Xena Browser [112]. Both studies reported better accuracy, sensitivity, and specificity when using transfer learning compared to non-transfer learning approaches based on the original tabular data.

We also identified a study reusing a model across species, which highlights another potential of transfer learning. Seddiki et al. [105] reused a model developed on mass spectrometry data from animals to classify human mass spectrometry data. Here, the benefits of transfer learning are clear, as genetic data from animals are often easier to retrieve and share due to the lack of privacy issues. This can potentially provide new opportunities within translational clinical research by reusing scientific knowledge gained from animal studies.

Wardi et al. [108] used both a transfer learning approach and a non-transfer learning approach to predict septic shock in an emergency department setting based on data from electronic health records (EHR) like blood pressure, heart rate, temperature, saturation, and blood sample values. They showed that transfer learning outperformed the traditional machine learning model, especially when only a smaller fraction of data was used. Furthermore, the transfer learning model was externally validated with promising results, which supports its clinical utility. The study by Wardi et al. makes individual-specific predictions possible, which is a prerequisite before an AI-based tool can be implemented in everyday clinical practice to support decision making [1].

Another promising use of transfer learning was described in the study by Gao et al. [99]. The authors developed models for various prediction tasks using genetic data from a heterogeneous population with a focus on how to translate models that perform well on the ethnic majority group to ethnic minority groups. Transfer learning clearly improved performance as compared to other machine learning approaches developed from scratch separately for each ethnic minority group. This application demonstrates how transfer learning can be used to tackle inequalities in health research.

## Text

We identified only seven studies using transfer learning on text. Applications ranged from risk assessment of psychiatric stressors, diseases, and medication abuse [20, 113–115], to prediction of morbidity, mortality [116, 117], and adverse incidents from oncological radiation [118]. It is somewhat surprising that we found so few studies, given the massive interest in NLP and the field's rapid development in recent years [119, 120]. The main reason behind this is methodological. Our systematic search could only identify studies explicitly mentioning transfer learning or a similar term (e.g. 'knowledge transfer'), however this does not seem to be a common practice in medical text analysis [121–123]. Future studies are warranted about the impact of

specific models (e.g. BERT]) on clinical research. There are also technical challenges in medical text analysis (e.g. ambiguous abbreviations, specialty-specific jargon, etc.) [124]. We can indirectly observe this in our review, where all seven articles were written by clinicians and technical authors together or by clinicians alone, in contrast to other data types. This is not to say that technical authors are not interested in the use of transfer learning on clinical text, but rather a reflection that transfer learning in text analysis is still a relatively new method, and studies often focus on technical aspects of NLP and are published in technical journals [119, 120].

Another important reason why we found so few articles on text is that there currently only exist a few large, annotated, and publicly available datasets with EHR [124]. Removing patient identifiable data from EHR is one of the key challenges currently limiting the sharing of large medical text datasets, though automatic tools have recently been developed to fasten this task [125]. Indeed, among the seven identified studies, three analyzed public social media posts to predict mental illnesses, two used relatively small institutional EHR datasets (<1000 patients), one used a large UK database with restricted access, and only a single study used a large, curated, and openly available dataset (the MIMIC-III critical care database [126]). A review on the use of all types of NLP on radiological reports found a similar pattern, where most studies used institutional EHR datasets [127], again highlighting the need for more large, annotated, and publicly available datasets.

As a final note on transfer learning in text, we would like to highlight the study by Si et al. [117]. In this paper, the authors proposed a new method to analyze medical records to predict mortality and identify obesity-related comorbidities. In brief, this method took the temporal aspect of each patient's documents into account, when predicting the patient's risk of mortality within the next year. It makes intuitive sense from a clinical perspective that a recent myocardial infarction could be more informative for mortality than one more than ten years ago, and this technical development could be important for future research in clinical text analysis.

## Limitations

In our study protocol, we choose to exclude studies combining source and target data (n = 30) from our review, even though we acknowledge the value of this approach in certain scenarios (e.g. [128]). We felt, however, that current data sharing restrictions would present barriers to the practical use and clinical impact of this type of transfer learning. Fine-tuning and feature-representation transfer, on the other hand, can aid to overcome such barriers by reusing models instead of the actual data, which is why these two forms of transfer learning were in the focus of our review.

The identification of clinical studies turned out to be more challenging than we expected, as it is a concept that is difficult to define. We had long discussions on whether to include brain-computer interface studies [129], where transfer learning seems to be a promising method to speed up calibration, with the final aim of helping people with a clinical condition. However, we concluded that the actual prediction models solve engineering problems rather than predict a clinical outcome. Similarly, we excluded NLP studies on biomedical named entity recognition, because even though the datasets might have been medically relevant documents, the task of identifying specific terms are often indirectly relevant from a clinical perspective e.g. by speeding up knowledge synthesis. A recent review provides an overview of the approach and its clinical applications [130].

The type of the target data might also be ambiguous, depending on how we define raw data. Transcripts of audio recordings could easily be considered as text, but we tried to evaluate what the first dataset was that e.g. a medical device output without further processing. Using

this approach, the recordings were considered as a dataset of audio type and transcribing them to text was considered as a data transformation.

Our scoping review did not include clinical studies if they were only published as preprints, or in proceedings of computer science conferences. However, we chose our inclusion criteria this way consciously, so that we could give an overview of the field from the perspective of clinical researchers. Therefore, our review does likely not include the latest technical developments within AI research and transfer learning, but instead include the techniques which have started to impact clinical research.

During data extraction, we planned to find the best proxy variables to answer our research questions, but some of these were challenging. As described previously, we decided not to extract quantitative data on the size of the source and the target datasets, as the unit of observation often differed. It was also difficult to characterize comparisons of transfer learning vs. non-transfer learning solutions because many studies reported many different models and performance metrics. We were curious about the background of the authors (clinical vs. technical), but used only the affiliations as a proxy, as that was easily accessible information in most cases.

## Conclusions

Our scoping review is a roadmap to transfer learning for non-image data in clinical research, providing clinical researchers an easily accessible resource to this relatively new technique in machine learning. The interest in transfer learning for non-image data in clinical research began to increase rapidly only recently, lagging a few years behind trends in its use for medical image analysis. Applications are unbalanced between different clinical research areas and data types. Neurology and cardiology seem to be among the 'first movers' with time series data, partly driven by the public availability of EEG and ECG datasets, which also suit machine learning due to their high-resolution nature. We found fewer classical epidemiological studies than expected despite transfer learning can help to overcome some of the big challenges of the field i.e. data collection on a large scale is often difficult and expensive, and data sharing can be hindered by privacy concerns. In the future, some of the largest epidemiological datasets (e.g. UK Biobank [131]) could serve as source data for the development of machine learning algorithms, that a wide range of smaller studies could build on by using transfer learning without access to the actual dataset. This is in line with the FAIR principles [132] supporting the reuse of digital assets in an environment with increased volume and complexity of datasets. Moreover, the wider use of reproducible research principles and stronger interdisciplinary collaborations between clinical researchers and computer scientists are crucial for the development of clinically relevant prediction models that can be reused with transfer learning across studies, clinical specialties, or even species.

## Supporting information

**S1 PRISMA Checklist. PRISMA-ScR checklist.**
(PDF)

**S1 Data. Extracted data.**
(XLSX)

**S1 Code. Data analysis code.**
(PDF)

**S1 ICMJE Disclosure. Conflict of interest statements.**
(PDF)

**S1 Glossary. Machine learning and transfer learning glossary.**
(PDF)

## Acknowledgments

The authors are grateful to Anne Vils Møller (Librarian at the Royal Danish Library) for her valuable advice on the search strategy. We are also grateful for the comments of Professor Daniel R. Witte and the Epidemiology Group, Steno Diabetes Center Aarhus, Denmark, and of Zeinab Schäfer and Andreas Mathisen from the Department of Computer Science, Aarhus University, Denmark, on the first draft of the manuscript.

## Author Contributions

**Conceptualization:** Andreas Ebbehoj, Mette Østergaard Thunbo, Ole Emil Andersen, Michala Vilstrup Glindtvad, Adam Hulman.

**Data curation:** Andreas Ebbehoj, Mette Østergaard Thunbo, Ole Emil Andersen, Adam Hulman.

**Formal analysis:** Michala Vilstrup Glindtvad, Adam Hulman.

**Investigation:** Andreas Ebbehoj, Mette Østergaard Thunbo, Ole Emil Andersen, Adam Hulman.

**Methodology:** Andreas Ebbehoj, Mette Østergaard Thunbo, Adam Hulman.

**Project administration:** Adam Hulman.

**Resources:** Andreas Ebbehoj, Adam Hulman.

**Software:** Andreas Ebbehoj, Michala Vilstrup Glindtvad, Adam Hulman.

**Supervision:** Adam Hulman.

**Visualization:** Michala Vilstrup Glindtvad, Adam Hulman.

**Writing – original draft:** Andreas Ebbehoj, Mette Østergaard Thunbo, Ole Emil Andersen, Adam Hulman.

**Writing – review & editing:** Andreas Ebbehoj, Mette Østergaard Thunbo, Ole Emil Andersen, Michala Vilstrup Glindtvad, Adam Hulman.

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
