## [Decision Letter · Decision Letter 0]

29 Nov 2021

PDIG-D-21-00103

Transfer learning for non-image data in clinical research: a scoping review

PLOS Digital Health

Dear Dr. Hulman,

Thank you for submitting your manuscript to PLOS Digital Health. After careful consideration, we feel that it has merit but does not fully meet PLOS Digital Health’s publication criteria as it currently stands. Therefore, we invite you to submit a revised version of the manuscript that addresses the points raised during the review process.

We look forward to receiving your revised manuscript.

Kind regards,

Matthew Chua Chin Heng

Academic Editor

PLOS Digital Health

Journal Requirements:

1. We ask that a manuscript source file is provided at Revision. Please upload your manuscript file as a .doc, .docx, .rtf or .tex. If you are providing a .tex file, please upload it under the item type ‘LaTeX Source File’ and leave your .pdf version as the item type ‘Manuscript’.

2. Please provide separate figure files in .tif or .eps format only, and remove any figures embedded in your manuscript file. Please ensure that all files are under our size limit of 20MB. If you are using LaTeX, you do not need to remove embedded figures.

3. We have noticed that you have uploaded supporting information but you have not included a list of legends.  Please add a full list of legends for all supporting information files (including figures, table and data files) after the references list. 

4. We have noticed that you have uploaded raw data files as supporting information but you have not included them in the list of legends.  Please add a citation of this SI data files into the list of legends of your supporting information files.

Additional Editor Comments (if provided):

Reviewers' comments:

Reviewer's Responses to Questions

**Comments to the Author**

1. Does this manuscript meet PLOS Digital Health’s publication criteria? Is the manuscript technically sound, and do the data support the conclusions? The manuscript must describe methodologically and ethically rigorous research with conclusions that are appropriately drawn based on the data presented.

Reviewer #1: Yes

Reviewer #2: Yes

2. Has the statistical analysis been performed appropriately and rigorously?

Reviewer #1: Yes

Reviewer #2: Yes

3. Have the authors made all data underlying the findings in their manuscript fully available (please refer to the Data Availability Statement at the start of the manuscript PDF file)?

Reviewer #1: Yes

Reviewer #2: Yes

4. Is the manuscript presented in an intelligible fashion and written in standard English?

Reviewer #1: Yes

Reviewer #2: Yes

5. Review Comments to the Author

Reviewer #1: This manuscript provides a survey of the current utilization of transfer learning in clinical studies which focus on non-image applications. The authors focus on clinical journals rather than technical or interdisciplinary venues (e.g., "Machine Learning for Healthcare" (https://www.mlforhc.org/) is not included). In general, I think such a survey is a useful contribution and highlights several gaps/lags between the general machine learning and clinical research communities.

My primary concern with the manuscript in its current form is a potentially too restrictive set of inclusion criteria, which does not consider several (arguably valid) common use cases of transfer learning, particularly in the NLP/text modality.

Comments/Questions

"Finally, we also excluded studies which combined source data with the target data as an integral part of their analysis as opposed to reusing only a source model."

This strikes me as unnecessarily strict a requirement. Augmenting a source model in some fashion using source+target data is quite reasonable and is a beneficial application of transfer learning. Can the authors elaborate on their rational here?

The inclusion/exclusion criteria aren't entirely clear to me and for findings for studies using transfer learning for text ("text (n=7; 8%)") seem far too low.

For example, these papers are all findable by PubMed and seem, at first cut, to use transfer learning and pass the 5 inclusion criteria outlined on page 6 of the manuscript.

- "Classifying Alzheimer's Disease Using Audio and Text-Based Representations of Speech" (Jan 2021) https://pubmed.ncbi.nlm.nih.gov/33519651/

- "Inferring ADR causality by predicting the Naranjo Score from Clinical Notes" (Jan 2021) https://www.ncbi.nlm.nih.gov/pmc/articles/PMC8075501/

- "Exploring Fever of Unknown Origin Intelligent Diagnosis Based on Clinical Data: Model Development and Validation" (Nov 2020) https://pubmed.ncbi.nlm.nih.gov/33172835/

et alia.

These studies seems to reflect the criteria "be a clinical study or focus on clinically relevant outcomes or measurement". I don't quite buy the claim (page 22) that tasks like NER should be excluded since they "only indirectly relevant from a clinical perspective e.g. by speeding up knowledge synthesis." In many clinical analyses, patient outcomes are directly determined by looking for textual evidence in clinical notes, so any text mining technique that leverages some flavor of BERT to identify patient outcomes should arguably be included. Not including this common use case undercuts the manuscript's findings about the limited uptake of transfer learning for text/NLP.

The "discussion" section is quite long and I feel some content could be moved out of this section and into "results"

Reviewer #2: This paper reports a scoping review of transfer learning for non-image data in clinical research, discusses its trends and highlights numerous potential works.

Overall the review is well performed, with a comprehensive review of the current literature. The review is also well written, balanced and clear. This paper fills an essential gap in the current literature.

6. PLOS authors have the option to publish the peer review history of their article (what does this mean?). If published, this will include your full peer review and any attached files.

**Do you want your identity to be public for this peer review?** For information about this choice, including consent withdrawal, please see our Privacy Policy.

Reviewer #1: No

Reviewer #2: No

---

## [Editor Report · Decision Letter 1]

15 Dec 2021

Transfer learning for non-image data in clinical research: a scoping review

PDIG-D-21-00103R1

Dear Dr. Hulman,

We're pleased to inform you that your manuscript has been judged scientifically suitable for publication and will be formally accepted for publication once it meets all outstanding technical requirements. 

Within one week, you'll receive an e-mail detailing the required amendments. When these have been addressed, you'll receive a formal acceptance letter and your manuscript will be scheduled for publication. The journal will begin publishing content in early 2022.

An invoice for payment will follow shortly after the formal acceptance. To ensure an efficient process, please log into Editorial Manager at https://www.editorialmanager.com/pdig/ click the 'Update My Information' link at the top of the page, and double check that your user information is up-to-date. If you have any billing related questions, please contact our Author Billing department directly at authorbilling@plos.org.

Kind regards,

Matthew Chua Chin Heng

Academic Editor

PLOS Digital Health

Additional Editor Comments (optional):

the paper has addressed the concerns from reviewers and is now ready for publication.